# WHEN SIMPLICITY WINS: EFFICIENT KNOWLEDGE GRAPH GENERATION WITH SEQUENTIAL DECODERS

## ABSTRACT

Knowledge Graph (KG) generation requires models to learn complex semantic dependencies between triples while maintaining domain validity constraints. State-of-the-art graph generation models rely on expensive attention mechanisms to capture complex dependencies, yet (head, relation, tail) triples can be straightforwardly represented as sequences, suggesting simpler architectures may suffice for KGs. We present **ARK** (**A**uto-**R**egressive **K**nowledge Graph Generation), a family of RNN and transformer-based models that succesfully perform KG generation. We show that the RNN variant requires only 9-21% of the training time, a 3.7-11× speedup, compared to the transformer variant. The RNN generates semantically valid graphs with 89.2-100.0% validity on IntelliGraphs benchmarks, with less than 0.76% degradation compared to the transformer on real-world datasets, while achieving up to 10.7% better compression rates on synthetic datasets and 12.8–21.1% gains on real-world datasets. Our analysis reveals that for KG generation, model capacity (hidden dimensionality $\geq 64$) matters more than depth, with single-layer GRUs matching deep transformer performance. We also introduce **SAIL**, an extension of ARK that adds variational latent variables for controlled diversity and interpolation in KG generation. Both models support unconditional sampling and conditional generation from partial graphs. Our findings challenge the assumption that structured data generation requires attention mechanisms. This efficiency gain can enable the generation of larger KGs and unlock new applications.

## 1 INTRODUCTION

Knowledge Graphs (KGs) encode knowledge as graphs of entities connected by typed relations, powering applications from search engines to drug discovery (Hogan et al., 2021). However, even large-scale KGs such as Wikidata miss substantial world knowledge. Although Knowledge Graph Embedding (KGE) models address incompleteness (Bordes et al., 2013; Yang et al., 2015), they score each triple independently, failing to capture the interdependencies that define valid knowledge structures. This independence assumption becomes particularly problematic for complex facts requiring multiple related triples to represent accurately (Nathani et al., 2019).

Consider representing "*Barack Obama was the US President from 2009-2017*" in a KG; this requires multiple interdependent triples that must also satisfy temporal constraints (start year $\leq$ end year), and type consistencies (only persons can be presidents). Traditional link predictors cannot ensure that these constraints are satisfied collectively, leading to semantically invalid predictions that undermine downstream reasoning tasks (Thanapalasingam et al., 2023). This is particularly critical for $N$-ary relations that inherently cannot decompose into independent binary predictions (Wen et al., 2016). In contrast to link prediction, KG generation addresses these limitations by modeling joint distributions over sets of triples, enabling the sampling of complete graphs that satisfy semantic constraints across all their components simultaneously.

Generative models can learn these interdependencies by modeling entire (sub)graphs rather than individual links (Xie et al., 2022). Previous work on generative models in the KG domain has primarily focused on generating triples from text (Saxena et al., 2022; Chen et al., 2020) or learning embeddings for downstream tasks (Xiao et al., 2016; He et al., 2015), rather than learning distributions over complete graph structures. To our knowledge, no prior work has demonstrated the ability

to sample entire, semantically valid KGs from learned probabilistic models. This raises a fundamental question: *What is actually required to model $p_\theta(G)$?* Recent generative approaches in the domain of simple graphs rely on computationally expensive transformer architectures for structured output generation (Yun et al., 2019; Zhuo et al., 2025; Zhao et al., 2025), raising a further question *Are computationally costly attention mechanisms really required for graph generation?*

We observe that Knowledge Graphs are naturally represented as sequences of triples (head, relation, tail), making them amenable to sequential generation. We introduce two autoregressive models: ARK, a GRU-based model, and $t$-ARK, a transformer-based model. Both generate KGs with 89.2-100.0% validity across diverse datasets. We further present **SAIL** (**S**equential **A**uto-Regress**I**ve Knowledge Graph Generation with **L**atents), a light-weight probabilistic extension of ARK that enables controllable generation from learned latent distributions.

Our contributions are as follows:

1. We introduce generative models based on RNNs and transformers that successfully solve the IntelliGraphs benchmark, achieving 99-100% semantic validity across diverse KG generation tasks, establishing that complex KG generation with semantic constraints is achievable through learned models;

2. We present two architectures: ARK, a lightweight GRU-based decoder, and SAIL, its extension with a learnable latent space representation enabling controlled generation, both capable of unconditional and conditional graph generation. We release our models and code on https://anonymous.4open.science/r/ARK-232F;

3. We demonstrate that recurrent architectures often outperform transformer-based alternatives, achieving comparable semantic validity while requiring only 9-21% of the training time and delivering 12.8-21.1% better prediction (in bits per graph) on real-world datasets;

4. We provide a comprehensive analysis of ARK and SAIL, showing that model capacity ($d \geq 64$) matters more than depth for KG generation, with single-layer GRUs matching deep transformers, with more interpretable latent structures that better capture semantics.

## 2 PRELIMINARIES

**Knowledge Graph Generation** We consider the task of generating semantically valid Knowledge Graphs $G = (E, R, T)$ where $E$ is a set of entities, $R$ is a set of relations, and $T \subseteq E \times R \times E$ is a set of triples. Unlike link prediction, which focuses on individual triple classification, our goal is to generate a collection of triples (*i.e.* subgraphs) that satisfy domain-specific semantic constraints while capturing interdependencies. This subgraph inference task is particularly crucial for $N$-ary relations and more complex facts that cannot decompose into independent binary predictions (Thanapalasingam et al., 2023). For example, temporal constraints require that *start year* precede *end year* across multiple triples, while type constraints ensure that only valid entity-relation combinations appear together. Models must generate and validate entire structures collectively rather than scoring individual triples. This is distinct from link prediction or generation of triples from text, as the models need to assign probability to and sample entire sets of interdependent triples.

**Definition 2.1** (Knowledge Graph Generation). Given a training set of Knowledge Graphs $\mathcal{D} = \{G_1, ..., G_n\}$, learn a generative model $p_\theta(G)$ that can sample new graphs $G' \sim p_\theta$ such that $G'$ that satisfies semantic validity constraints $\mathcal{S}$ while not appearing in $\mathcal{D}$.

**Definition 2.2** (Semantic Validity). A graph $G$ is semantically valid if it satisfies constraints $\mathcal{S} = \{s_1, ..., s_k\}$ where each $s_i$ is a rule (e.g., type constraints, temporal consistency, relational dependencies). For example, in syn-tipr: $s_1$: `start_year` $\leq$ `end_year`.

**Variational Inference** To learn latent representations, we use the $\beta$-VAE framework (Kingma & Welling, 2013; Higgins et al., 2017), which aims to maximize the evidence lower bound (ELBO):

$$\mathcal{L}(\phi, \theta; G) = \mathbb{E}_{q_\phi(z|G)}[\log p_\theta(G|z)] - \beta \, \mathrm{KL}[q_\phi(z|G)||p(z)]. \tag{1}$$

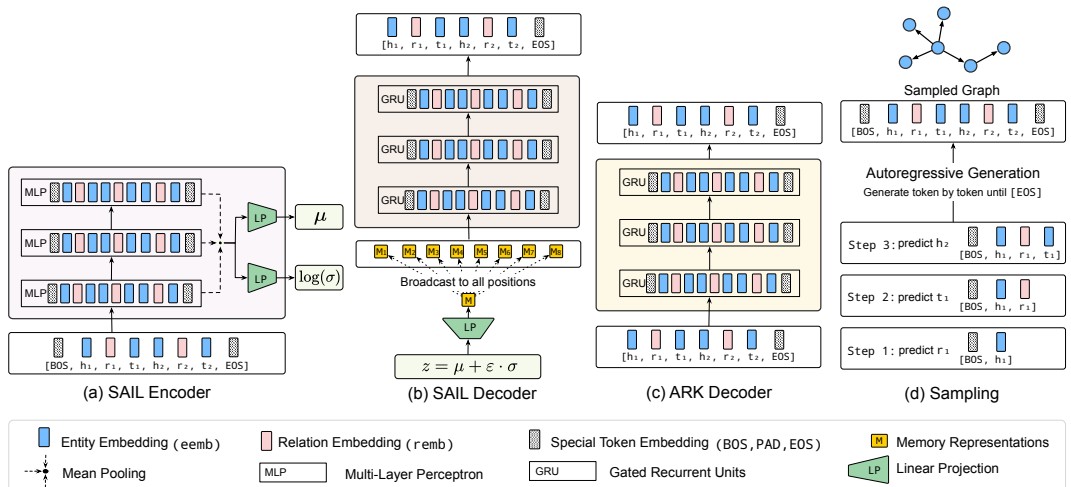

Figure 3.1: Overview of Model Architectures. **(a) SAIL Encoder:** Multi-layer perceptron (MLP) processes linearized KG sequences $[\texttt{BOS}, h_1, r_1, t_1, h_2, r_2, t_2, \ldots, \texttt{EOS}]$, with mean pooling to produce fixed-size representations. Linear projections generate latent distribution parameters $\mu$ and $\log \sigma$. **(b) SAIL Decoder:** GRU-based decoder conditions on sampled latent code $z \sim \mathcal{N}(\mu, \sigma^2)$ by broadcasting $z$ to all sequence positions and concatenating with embeddings $[M_1, M_2, \ldots, M_n]$ at each timestep. **(c) ARK Decoder:** GRU decoder for ARK operates without latent conditioning, processing embedded sequences directly through stacked GRU layers. **(d) Sampling:** Autoregressive generation proceeds token-by-token with causal masking until $\texttt{EOS}$ token or maximum length.

## 3 SEQUENTIAL DECODING FOR KNOWLEDGE GRAPH GENERATION

We present our approach to Knowledge Graph generation through sequential decoding. We first describe how graphs are processed as input sequences, then introduce ARK (**A**uto-**R**egressive **K**nowledge Graph Generation), our GRU-based decoder model, followed by SAIL (**S**equential Probabilistic **A**uto-**R**egressive **K**nowledge Graph Generation), which extends ARK with a variational framework for probabilistic generation.

### 3.1 GRAPH INPUT PROCESSING

To enable sequential generation, we linearize KGs into token sequences. A graph $G$ containing triples $(h_1, r_1, t_1), ..., (h_n, r_n, t_n)$ is represented as $[\texttt{BOS}, h_1, r_1, t_1, h_2, r_2, t_2, ..., h_n, r_n, t_n, \texttt{EOS}]$, where $\texttt{BOS}$ marks the sequence start and $\texttt{EOS}$ indicates termination. These tokens provide explicit generation boundaries, enabling the model to learn proper initiation and termination conditions. We employ a unified vocabulary $\mathcal{V} = \{\texttt{BOS}, \texttt{PAD}, \texttt{EOS}\} \cup \mathcal{E} \cup \mathcal{R}$ that combines special tokens, entities, and relations into a single embedding space. Variable-length graphs are padded to a fixed maximum length $L_{\max}$ using $\texttt{PAD}$ tokens for batched training. During training, triple ordering within sequences is randomized to prevent the model from learning spurious positional patterns.[1]

### 3.2 AUTOREGRESSIVE KNOWLEDGE GENERATION (ARK)

ARK is an autoregressive RNN model (Mikolov et al., 2010), using Gated Recurrent Units (GRUs) (Cho et al., 2014). Given a sequence representation, the model learns to predict the next token conditioned on all previous tokens, exploiting the natural sequential structure of linearized graphs. See Appendix A.3.1 for details.

The model is trained autoregressively with cross-entropy loss, conditioning on ground-truth previous tokens: $\mathcal{L}_{\text{ARK}} = -\sum_{t=1}^{T} \log p(x_t | x_{<t})$.

---

[1]This prevents leakage. *e.g.* in syn-paths, linearizing in path order makes the pattern easier to learn.

**Generation**    During inference, ARK generates graphs sequentially starting from the BOS token. At each timestep $t$, the model computes the probability distribution $p(x_{t+1}|x_{\leq t})$ over the vocabulary. We select the next token through sampling controlled by temperature and top-$k$. Concretely, we divide logits by temperature $T$, keep only the top-k tokens, then retain the smallest prefix whose cumulative probability mass exceeds $p$ (top-$p$), renormalize and sample one token. Decoding stops on EOS or when the maximum graph length has been reached. The generated sequence is parsed into triples by extracting consecutive $(h, r, t)$ token triplets between BOS and EOS markers, with incomplete triples discarded during post-processing.

### 3.3 Sequential Autoregressive Knowledge Graph Generation with Latents (SAIL)

SAIL extends ARK by incorporating a variational autoencoder framework, similar to Bowman et al. (2016), enabling probabilistic generation from learned latent distributions, $z$. This extension allows for controlled generation and interpolation in latent space while maintaining the efficiency of GRU-based decoding.

**Encoder**    The encoder processes the input sequence through a multi-layer perceptron (MLP) to obtain a fixed-size representation. Each input triple $(h, r, t)$ is embedded as $[E_e[h]; E_r[r]; E_e[t]] \in \mathbb{R}^{3d}$, and we take a mean over the sequence to form a graph-level vector. The MLP consists of multiple dense layers, with the number of these layers chosen to match the number of stacked GRU layers in the decoder. The encoder then processes the aforementioned sequence through these layers, while ReLU is used as the activation function. The final hidden representation is projected to latent distribution parameters, $\boldsymbol{\mu}$ and $\log \boldsymbol{\sigma}^2$.

**Latent Sampling**    We sample from the latent distribution using the reparameterization trick:

$$\mathbf{z} = \boldsymbol{\mu} + \boldsymbol{\sigma} \odot \boldsymbol{\epsilon}, \quad \boldsymbol{\epsilon} \sim \mathcal{N}(0, \mathbf{I}) \tag{2}$$

**Decoder**    The decoder extends ARK's GRU architecture by conditioning on the latent variables, $\mathbf{z}$. The latent representation is first projected and used to initialize the decoder's hidden state: $\mathbf{h}_0 = \tanh(\mathbf{W}_{\text{init}}\mathbf{z} + \mathbf{b}_{\text{init}})$ To maintain global conditioning throughout generations, $\mathbf{z}$ is broadcast to all sequence positions. At each timestep, we concatenate the projected latent code with the input embedding: $\mathbf{x}'_t = [\mathbf{x}_t; \mathbf{W}_z\mathbf{z}]$ This ensures that the global graph structure encoded in $\mathbf{z}$ influences every token prediction, allowing the decoder to maintain semantic consistency across the entire sequence. SAIL is trained by maximizing the ELBO (as shown in Equation 1).

**Generation & Sampling**    To generate a graph using the model, we sample $\mathbf{z} \sim \mathcal{N}(0, \mathbf{I})$ from the prior distribution. We call this *unconditional generation*. Additionally, we define *conditional generation* where we encode a partial graph to obtain the posterior $q(\mathbf{z}|G_{\text{partial}})$, sample from it, and then complete the sequence. The generation then follows an autoregressive process where the probability of the complete graph factorizes as: $p_\theta(G|\mathbf{z}) = \prod_{t=1}^{T} p_\theta(x_t|x_{<t}, \mathbf{z})$. We use beam search with $\text{score}(x_{1:t}|\mathbf{z}) = \sum_{i=1}^{t} \log p_\theta(x_i|x_{<i}, \mathbf{z})$. Latent conditioning enables controlled generation by manipulating $\mathbf{z}$, we can interpolate between graphs or explore specific regions of the latent space to generate graphs with desired properties.

## 4 Evaluation

We evaluate a family of RNN and transformer-based models on the IntelliGraphs benchmark (Thanapalasingam et al., 2023), which consists of five datasets designed to test different aspects of Knowledge Graph generation.

**Benchmark**    IntelliGraphs includes three synthetic datasets (syn-paths, syn-types, syn-tipr) with algorithmically verifiable semantics, ranging from simple path structures to temporal constraints requiring reasoning about time intervals, and two real-world Wikidata-derived datasets (wd-movies, wd-articles) capturing complex relational patterns from movie and academic publication domains. Synthetic datasets contain fixed-size graphs (3-5 triples) with small vocabularies (30-130 entities), while Wikidata datasets feature variable-size graphs (2-212 triples) with large entity vocabularies (24K-61K entities), providing diverse challenges for evaluating generation quality and semantic validity. Detailed dataset characteristics and semantic constraints are provided in Appendix A.2. To

the best of our knowledge, IntelliGraphs is the only benchmark for KG generation, while other KG datasets focus on link prediction.

**Baselines** The probabilistic baselines from Thanapalasingam et al. (2023) decompose graph generation as $p(F) = p(S|E)p(E)$, where $E$ represents entities and $S$ represents structure. The *uniform* baseline samples from uniform distributions, providing estimates for compression bits by assuming equal likelihood for all configurations. The KGE-based baselines (TransE, ComplEx, DistMult) estimate $p(E)$ using entity frequencies with Laplace smoothing and $p(S|E)$ using learned scoring functions: TransE models relations as translations (Bordes et al., 2013), DistMult uses bilinear interactions (Yang et al., 2015), and ComplEx employs complex-valued embeddings (Trouillon et al., 2016). These models convert scores to probabilities through sigmoid functions and compute compression as $-\log_2 p(S|E) - \log_2 p(E)$. To enable fair architectural comparison, we introduce transformer-based variants: $t$-ARK replaces ARK's GRU decoder with a transformer decoder using causal self-attention for capturing long-range dependencies, while $t$-SAIL (depicted in Figure C.1 in the Appendix) extends this with a variational framework employing transformer encoders and decoders throughout. These transformer variants offer potentially superior modeling of complex patterns, but require substantially more computational resources than their GRU counterparts.

**Evaluation Metrics** We evaluate generation quality through three primary metrics: (1) *Semantic Validity* – the proportion of generated graphs that satisfy dataset-specific semantic constraints, measuring whether the model learns to respect domain rules without explicit supervision; (2) *Novelty* – the proportion of generated graphs not present in the training set, distinguishing genuine generation from memorization; and (3) *Compression* – the information-theoretic measure $-\log p(G)$ in bits, quantifying how efficiently the model encodes graph structure. For variational models, we additionally report the KL divergence between the approximate posterior and prior. These metrics collectively assess whether models capture the underlying data distribution while maintaining semantic coherence and generalization capability.

### 4.1 COMPRESSION CODE LENGTH

We express the negative log-likelihood, $-\log_2(p_\theta)$, in bits-per-graph. See Appendix A.3.2 for details. This measures both the ability to compress and to predict (Grünwald, 2007, Section 3.2).

**Results** Table 1 shows the compression performance across all models. The decoder-only models (ARK and $t$-ARK) achieve competitive compression rates, particularly excelling on synthetic datasets with 27.65 and 27.57 bits for syn-paths (outperforming the uniform baseline of 30.49 bits) and exceptional performance on syn-tipr (23.48 and 23.34 bits respectively). While their compression on syn-types is higher (59.63 and 59.79 bits), both models compensate with strong semantic validity in generation tasks. On real-world datasets, ARK achieves the best overall compression with 98.19 bits for wd-movies and 205.24 bits for wd-articles, demonstrating efficient encoding despite increased graph complexity. In contrast, the variational models (SAIL and $t$-SAIL) report ELBO upper bounds rather than exact compression, as they use latent vectors $\mathbf{z}$ to capture graph structure. Their compression includes both reconstruction and KL divergence terms, with the KL component varying from nearly zero to syn-types (0.15 bits) to moderate values on other datasets (13-32 bits), indicating adaptive latent space usage on dataset complexity.

### 4.2 SAMPLING FROM LATENT VARIABLE, $z$

We assess the generative capabilities of SAIL through two complementary approaches: unconditional generation by sampling from the prior distribution $p_\theta(z)$, and conditional generation by providing partial graph sequences. These experiments test whether the learned latent space is well-structured and whether the model can generate semantically valid, novel graphs, demonstrating true generative modeling rather than mere memorization. For more details regarding the method and qualitative analysis, we refer the reader to Appendices A.3.3 and A.4, respectively.

**Quantitative Results** Table 1 shows unconditional graph generation results. ARK achieves exceptional semantic validity across synthetic datasets, with 99.95% on syn-paths, 100.00% on syn-tipr, and 89.22% on syn-types. SAIL similarly demonstrates strong performance with 92.50%, 98.45%, and 100.00% validity, respectively. Both dramatically outperform KGE baselines (TransE, DistMult, ComplEx), which achieve less than 1% validity and produce 76–100% empty graphs,

| Datasets | Model | % Valid Graphs ↑ | % Novel & Valid ↑ | % Empty Graphs ↓ | Compression Length (bits) ↓ |
|---|---|---|---|---|---|
| **syn-paths** | uniform | 0 | 0 | 0 | 30.49 |
| | TransE | 0.25 | 0.25 | 76.55 | 49.89 |
| | DistMult | 0.69 | 0.69 | 85.41 | 54.39 |
| | ComplEx | 0.71 | 0.71 | 85.73 | 48.58 |
| | $t$-SAIL | 99.60 | 99.60 | 0 | 27.77 |
| | SAIL | 92.50 | 92.50 | 0 | 28.74 |
| | $t$-ARK | 97.39 | 97.39 | 0 | **27.57** |
| | ARK | **99.95** | **99.95** | 0 | 27.65 |
| **syn-tipr** | uniform | 0 | 0 | 0 | 61.61 |
| | TransE | 0 | 0 | 94.42 | 69.51 |
| | DistMult | 0 | 0 | 86.66 | 63.96 |
| | ComplEx | 0 | 0 | 96.05 | 67.51 |
| | $t$-SAIL | 100.00 | 100.00 | 0 | 26.30 |
| | SAIL | 98.45 | 98.45 | 0 | 27.14 |
| | $t$-ARK | 100.00 | 100.00 | 0 | **23.34** |
| | ARK | **100.00** | **100.00** | 0 | 23.48 |
| **syn-types** | uniform | 0 | 0 | 0 | **36.02** |
| | TransE | 0.21 | 0.21 | 84.56 | 48.26 |
| | DistMult | 0.13 | 0.13 | 87.53 | 47.69 |
| | ComplEx | 0.07 | 0.07 | 89.75 | 47.69 |
| | $t$-SAIL | **100.00** | **100.00** | 0 | 59.61 |
| | SAIL | 100.00 | 100.00 | 0 | 60.58 |
| | $t$-ARK | 87.07 | 87.07 | 0 | 59.79 |
| | ARK | 89.22 | 89.22 | 0 | 59.63 |
| **wd-movies** | uniform | 0 | 0 | 0 | 171.60 |
| | TransE | 0 | 0 | 85.39 | 208.60 |
| | DistMult | 0 | 0 | 87.07 | 202.68 |
| | ComplEx | 0 | 0 | 98.13 | 208.50 |
| | $t$-SAIL | **99.83** | **99.83** | 0 | 124.50 |
| | SAIL | 99.47 | 99.47 | 0 | 116.84 |
| | $t$-ARK | 98.33 | 98.33 | 0 | 114.49 |
| | ARK | 99.24 | 99.24 | 0 | **98.19** |
| **wd-articles** | uniform | 0 | 0 | 0 | 693.80 |
| | TransE | 0 | 0 | 95.42 | 910.65 |
| | DistMult | 0 | 0 | 100.00 | 887.30 |
| | ComplEx | 0 | 0 | 97.54 | 901.91 |
| | $t$-SAIL | 98.00 | 98.00 | 0 | 235.24 |
| | SAIL | **99.13** | **99.13** | 0 | **199.55** |
| | $t$-ARK | 95.37 | 95.37 | 0 | 224.25 |
| | ARK | 97.24 | 97.24 | 0 | 205.24 |

Table 1: Semantic validity and compression length in bits of the graphs generated. We sample graphs and check the novelty of the sampled graphs by comparing them against the training and validation sets. We use the test set for the calculation of the compression length when training has finished. The best performing models for each dataset are **bolded**. Baseline results are from the IntelliGraphs paper (Thanapalasingam et al., 2023). The full results are available in Tables 4 and 5 in the Appendix.

suggesting they fail to learn meaningful latent representations. All generated graphs from our models are novel rather than memorizing training examples. For real-world datasets, ARK maintains 99.24% validity on wd-movies and 97.24% on wd-articles, while SAIL achieves 99.47% and 99.13% respectively, demonstrating robust performance despite increased complexity.

## 4.3 INTERPOLATION IN LATENT SPACE

For SAIL and $t$-SAIL, we explore the structure of the learned latent space by interpolating between encoded representations of different graphs. This analysis reveals whether the model learns smooth, semantically meaningful transitions between graph structures, indicating a well-organized

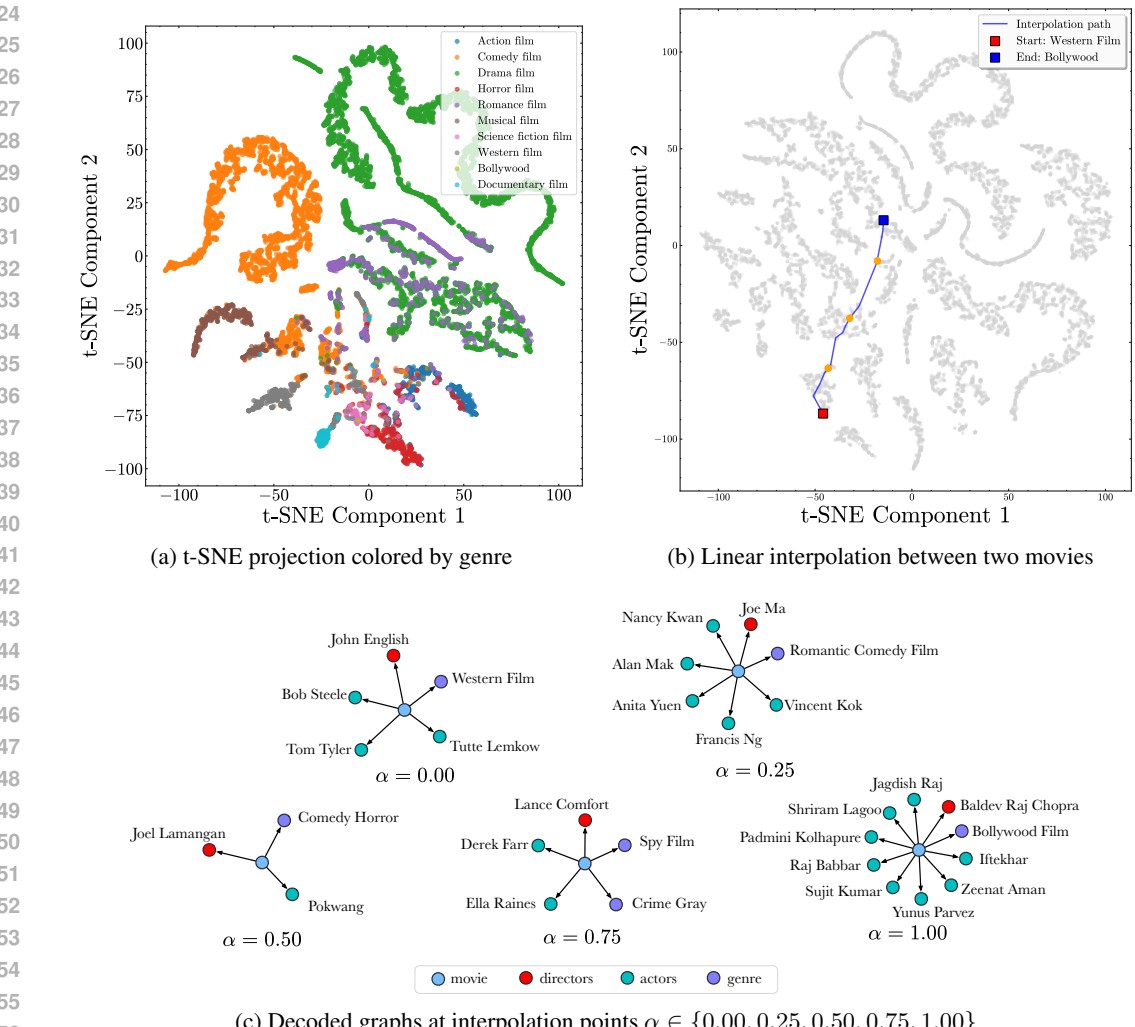

(a) t-SNE projection colored by genre

(b) Linear interpolation between two movies

(c) Decoded graphs at interpolation points $\alpha \in \{0.00, 0.25, 0.50, 0.75, 1.00\}$

Figure 4.1: Latent space visualization for the wd-movies dataset. (a) t-SNE projection shows clear clustering by genre. (b) Smooth interpolation paths connect different movie types. (c) Decoded graphs along the interpolation path show gradual transitions in cast and genre attributes, maintaining semantic validity throughout.

latent space where similar graphs cluster together and intermediate points correspond to valid hybrid structures. For more details regarding the method, we refer the reader to Appendix A.3.4.

**Quantitative Results** The smoothness metrics reveal distinct patterns across dataset complexity. Comparing SAIL and $t$-SAIL (Figure 6 in Appendix), we observe that $t$-SAIL generally achieves better latent space organization. For syn-tipr, $t$-SAIL shows exceptional quality with near-perfect local smoothness (0.99) and global consistency (0.98), while SAIL achieves lower but still strong metrics (0.93 and 0.69, respectively). The architectural difference is most pronounced on syn-paths, where $t$-SAIL maintains moderate global consistency (0.36) compared to SAIL's much weaker performance (0.14), suggesting that transformer-based encoders better capture graph structure. Surprisingly, SAIL demonstrates superior performance on syn-types with high local smoothness (0.92) and global consistency (0.73), exceeding $t$-SAIL's metrics (0.82 and 0.60). The flip rates reveal interesting trade-offs: SAIL shows higher instability on most datasets (0.33 for syn-paths, 0.40 for wd-movies) compared to $t$-SAIL (0.20 and 0.15 respectively), though they achieve similar rates on syn-tipr (0.10 vs. 0.09). Real-world datasets (wd-movies) show both models achieving strong local smoothness (0.84 and 0.87) but with $t$-SAIL maintaining better global consistency (0.58 vs 0.49). These results suggest that while transformer encoders generally provide better latent space orga-

nization, the simpler GRU-based SAIL can match or exceed transformer performance on certain structured datasets, particularly syn-types.

**Qualitative Results**  Figure 4.1 demonstrates the learned latent space structure for the wd-movies dataset. The t-SNE projection (Figure 4.1 a) reveals distinct clustering by genre, indicating that SAIL learns to organize its latent space according to semantic film categories without explicit supervision. The linear interpolation experiment (Figure 4.1 b,c) traces a path between a Western film and a Romantic Comedy, with decoded graphs at intermediate points ($\alpha \in \{0, 0.25, 0.5, 0.75, 1\}$) demonstrating smooth transitions: starting from a Western with actors Bob Steele and Tom Tyler, progressing through hybrid representations with mixed genre elements (Comedy Horror at $\alpha = 0.50$), and reaching a Thriller film with different cast members. While all intermediate graphs maintain a valid KG structure, the semantic coherence varies; intermediate points produce valid but potentially less realistic combinations of actors and genres, suggesting that semantic validity is preserved throughout interpolation, but semantic plausibility is highest near the training data manifold, consistent with typical VAE behavior on structured data.

### 4.4 ABLATION STUDY

The ablation study in Appendix A.5 demonstrates that model capacity (hidden dimensionality) is more critical than network depth, with a clear performance threshold at $d_{\mathrm{model}} = 64$ for the syn-paths dataset. Simpler architectures, such as GRU decoders, achieve comparable validity and novelty rates while providing substantial training speedups. Compression efficiency improves with model complexity, yet lightweight architectures match transformer-based models in generation quality.

## 5 RELATED WORK

While substantial progress has been made in related areas such as molecular graph generation and graph neural networks, the unique challenges of KG generation, including semantic consistency, relational diversity, and logical constraints, have only begun to be addressed. Recent advances in graph representation learning have paved the way for developing generative modeling of KGs.

**Generative Modeling of KGs**  Cowen-Rivers et al. (2019) learn joint probability distributions over facts stored in Knowledge Graphs to estimate the predictive uncertainty of KGE models and evaluate their generative model using link prediction. TransG is a probabilistic model that learns the semantics of N-*ary* relations (Xiao et al., 2016). In contrast to this work, we focus on generating a collection of triples. Loconte *et al.* reinterprets the score functions of traditional KGEs as circuits, enabling efficient marginalization and sampling, thereby facilitating the generation of new triples consistent with existing KGs (Loconte et al., 2024). Notably, Galkin et al. (2024) proposes ULTRA, a foundation model for KG reasoning that achieves strong generalization across diverse KGs through a unified pre-training approach on multiple graphs, demonstrating that a single model can transfer reasoning capabilities across different knowledge domains without fine-tuning.

**Graph Transformers**  Machine learning on sets requires learning permutation-invariant functions (Zaheer et al., 2017). Various frameworks have been proposed that use attention mechanisms for graph representation learning (Kim et al., 2022; Yun et al., 2019; Zhuo et al., 2025; Zhao et al., 2025). Due to the fully-attentional nature of Transformers, they can be seen as a generalisation of Graph Neural Networks (Zaheer et al., 2017). In our work, we deal with a directed graph with labeled edges. Recent advances include GraphGPS (Rampášek et al., 2022), which combines message passing with global attention mechanisms, and NodeFormer (Wu et al., 2022), which efficiently computes all-pair interactions through kernelized softmax. Shirzad et al. (2023) introduce Exphormer, achieving linear complexity in graph transformers through virtual global nodes and expander graphs. Unlike these architectures that focus on encoding existing graphs, our work demonstrates that simpler recurrent architectures suffice for generating KGs, achieving comparable performance with 3.7-11x speedup over transformer-based generation models.

**Graph Generative Models**  Deep graph generative models have predominantly focused on generating novel molecular structures, emphasizing chemical validity and stability (Li et al., 2018). Beyond molecular applications, models like GraphVAE and GraphRNN have been developed to capture complex graph structures through latent variable and autoregressive approaches. Kipf et al. (2020) introduced methods to infer symbolic abstractions from visual data and relational structures

from observations. Recent developments include DiGress (Vignac et al., 2023), which applies discrete denoising diffusion to graph generation, and GraphARM (Kong et al., 2023), which combines autoregressive models with graph neural networks for scalable generation. Liu et al. (2024) propose GraphMaker, a diffusion-based approach that generates graphs by iteratively refining node features and edge structures. Our work extends these concepts by focusing on the semantic generation of KGs, learning implicit semantic constraints from background information without predefined rules.

**Neuro-Symbolic Generative Models for KGs** Combining distributed and symbolic representations, neuro-symbolic systems aim to combine the strengths of both paradigms (van Bekkum et al., 2021). Generative neuro-symbolic machine combines distributed and symbolic entity-based representations in a generative latent variable model to infer object-centric symbolic representations from images Jiang & Ahn (2020). Balloccu et al. (2024) introduces KGGLM, a generative language model designed for generalizable KG representation learning in recommender systems, which exemplifies the integration of neural and symbolic approaches. A recent breakthrough comes from van Krieken et al. (2025), who introduce a neurosymbolic diffusion model that integrates logical constraints directly into the diffusion process. While these approaches explicitly incorporate symbolic reasoning into neural networks, our work shows that simple recurrent models can implicitly learn and enforce semantic constraints through autoregressiveive generation, achieving 89.2-100.0% semantic validity without explicit logical rule integration.

# 6 CONCLUSION

We have demonstrated that KG generation can be performed effectively by sequential, autoregressive models and that lightweight GRU-based decoders can match and sometimes even exceed transformer performance while requiring only 9-21% of the training time: a 3.7-11× speedup. Our hypothesis is that the natural sequential representation of KGs, with facts presented as ordered (head, relation, tail) triples, aligns well with the inductive biases of recurrent architectures, enabling them to capture both local dependencies within triples and global semantic constraints across subgraphs. Unlike traditional KGE approaches that treat triples independently, ARK and SAIL preserve semantic validity by learning interdependencies, achieving 89.2-100.0% validity across diverse datasets while maintaining competitive compression rates. We note that the complexity of GRUs is $O(nd^2)$ operations per sequence, compared to transformers' $O(n^2d)$ complexity, showing that the former is more efficient where sequence length dominates.

**Limitations** Our work assumes a fixed vocabulary of entities and relations known at training time, limiting applicability to open-world scenarios where new entities emerge dynamically. While ARK and SAIL generate semantically valid graphs, in our experiments, we only test on relatively small graphs (3-212 triples). Additionally, the autoregressive formulation imposes a linear ordering on inherently unordered graph structures, though our experiments show that this does not significantly impact generation quality.

**Future Work** Several directions merit exploration: extending the ARK framework to handle *out-of-vocabulary* entities and relations through compositional embeddings or meta-learning approaches, investigating hierarchical generation strategies for larger graphs where local subgraphs are generated independently then composed, and making the learned semantic rules explicit rather than leaving them implicit in the model parameter would help to identify and mitigate learning undesired constraints that may stem from biases in the data. Since this work solves most challenges in the Intelligraphs benchmark, larger and more complex KG generation benchmarks are called for.

**Ethics Statement**    Datasets on which our models are trained may contain societal biases and factual errors, which could propagate through the learning process and manifest in generated knowledge graphs. While our models achieve high semantic validity scores, they may still reproduce or amplify biases present in the training data, potentially generating graphs that reflect historical inequities or stereotypes. Additionally, the autoregressive generation process could produce factually incorrect but semantically valid triples, as the model learns logical rules rather than verifying the truth. We intend for ARK and SAIL to be treated as research prototypes to advance the field of KG generation, and should not be deployed in critical applications without thorough testing and safeguards. See Thanapalasingam et al. (2023) for a detailed analysis of the limitations of the datasets.

**Reproducibility Statement**    We provide complete code and detailed configurations to ensure complete reproducibility of all experiments. Our implementation, including model architectures, training scripts, data preprocessing pipelines, and evaluation metrics, is available at `https://anonymous.4open.science/r/ARK-232F`. Experimental details, including hyperparameters, hardware specifications, and training procedures, are provided in Appendix A.1. We also release pre-trained model checkpoints for both ARK and SAIL to facilitate reproduction of our results and enable further research building upon our work. Detailed instructions for replicating each experiment, including expected runtimes and resource requirements, are provided in the repository.

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
