# A APPENDIX

## A.1 EXPERIMENTAL DETAILS

We used the PyTorch library [2] to develop and test the models. All experiments were performed on a single-node machine with an Intel(R) Xeon(R) Gold 5118 (2.30GHz, 12 cores) CPU and 64GB of RAM, with a single NVIDIA A100 GPU (80GB of VRAM) or a single NVIDIA H100 GPU (80GB of VRAM). We used PyTorch's CUDA acceleration for model training and inference. We used the Adam optimizer with variable learning rates (Kingma & Ba, 2015). We monitored the training of the models using the Weights & Biases package [3]. All experiments use the same train/validation/test splits as the original IntelliGraphs benchmark (Thanapalasingam et al., 2023) to ensure fair comparison.

**Use of Large Language Models** We used LLMs as assistive tools for writing refinement (*e.g.* grammar correction, sentence clarity), coding support (*e.g.*debugging PyTorch implementations, implementing standard metrics, and plotting graphs), and AI-enhanced search engines for literature discovery. These were verified for factual accuracy. All scientific contributions, from research ideation to experimental design and results interpretation, present original work by the authors.

**Hyperparameter Optimization** For ARK and SAIL, the hyperparameters were automatically tuned using grid search {learning rate, batch size, number of epochs, latent dimension size [4], number of neurons and number of layers[5] } to get the best performance for the validation split. For reproducibility, we provide an extension description of the hyperparameters as YAML files under the *configs* directory on https://anonymous.4open.science/r/ARK-232F.

## A.2 DATASET DETAILS

The IntelliGraphs benchmark datasets test different aspects of semantic validity and structural complexity:

1. **syn-paths:** A synthetic dataset containing path graphs with simple semantics that can be algorithmically verified in linear time. These are acyclic graphs where edge directions follow the path structure.

2. **syn-types:** A synthetic dataset featuring typed entities and relations where type constraints on entities depend on the relation type, enforcing semantic consistency through type checking.

3. **syn-tipr:** A synthetic dataset containing subgraphs based on the *Time-indexed Person Role* (tipr) ontology pattern.[6] The semantics are defined by the tipr graph pattern, requiring temporal reasoning to generate valid time intervals.

4. **wd-movies:** Small knowledge graphs describing movies, extracted from Wikidata.[7] Each graph contains one existential node representing the movie, with entity nodes for director(s) connected via `has_director`, cast members connected via `has_actor`, and genres connected via `has_genre` relations.

5. **wd-articles:** Small knowledge graphs that describe research articles, extracted from Wikidata. Each graph contains one existential node representing the article, with entity nodes for author(s) connected via `has_author`, publication venues connected via `published_in`, and topics connected via `has_topic` relations.

---

[2] https://pytorch.org/
[3] https://wandb.ai
[4] for SAIL only
[5] Both encoder's and decoder's neurons and number of layers. For models without encoder the tuning for the number of layers and neurons was done the decoder part
[6] http://ontologydesignpatterns.org/wiki/Submissions:Time_indexed_person_role
[7] https://www.wikidata.org

| Datasets | Dataset Size (Train/Val/Test) | Unique Entities | Relation Types | Triples per Graph |
|---|---|---|---|---|
| syn-paths | 60,000/20,000/20,000 | 49 | 3 | 3 |
| syn-types | 60,000/20,000/20,000 | 30 | 3 | 3 |
| syn-tipr | 50,000/10,000/10,000 | 130 | 5 | 5 |
| wd-movies | 38,267/15,698/15,796 | 24,093 | 3 | 2-23 |
| wd-articles | 54,163/22,922/22,915 | 60,932 | 6 | 4-212 |

Table 2: Dataset characteristics for the IntelliGraphs benchmark. Synthetic datasets (syn-*) have fixed graph structures while Wikidata-derived datasets (wd-*) exhibit variable sizes. Entity counts represent unique entities across all graphs; edge counts indicate the number of triples per individual graph.

## A.3 METHODS

Here, we provide more details about the methods we used for the empirical analyses of ARK and SAIL.

### A.3.1 GATED RECURRENT UNITS (GRUS)

The ARK model employs a standard GRU decoder with hidden state $\mathbf{h}_t \in \mathbb{R}^d$ that evolves as:

$$\mathbf{r}_t = \sigma(\mathbf{W}_r\mathbf{x}_t + \mathbf{U}_r\mathbf{h}_{t-1} + \mathbf{b}_r) \tag{3}$$

$$\mathbf{z}_t = \sigma(\mathbf{W}_z\mathbf{x}_t + \mathbf{U}_z\mathbf{h}_{t-1} + \mathbf{b}_z) \tag{4}$$

$$\tilde{\mathbf{h}}_t = \tanh(\mathbf{W}_h\mathbf{x}_t + \mathbf{U}_h(\mathbf{r}_t \odot \mathbf{h}_{t-1}) + \mathbf{b}_h) \tag{5}$$

$$\mathbf{h}_t = (1 - \mathbf{z}_t) \odot \mathbf{h}_{t-1} + \mathbf{z}_t \odot \tilde{\mathbf{h}}_t \tag{6}$$

where $\mathbf{r}_t$ and $\mathbf{z}_t$ are reset and update gates respectively, $\mathbf{x}_t$ is the embedding of the current input token, and $\odot$ denotes element-wise multiplication. At each timestep, the hidden state is projected to vocabulary logits: $p(x_{t+1}|x_{\leq t}) = \text{softmax}(\mathbf{W}_o\mathbf{h}_t + \mathbf{b}_o)$.

### A.3.2 COMPRESSION LENGTH

For both ARK and SAIL, we compute the compression length to generate graphs as sequences. Since ARK is a decoder-only autoregressive model, we compute:

$$\text{Compression Length of } G = -\log_2(p_\theta(G)) = -\sum_{t=1}^{T} \log_2(p_\theta(x_t|x_{<t})) \tag{7}$$

where $x_t$ represents the $t$-th token in the linearized graph sequence $[\text{BOS}, h_1, r_1, t_1, ..., \text{EOS}]$ and $T$ is the sequence length. Each term represents the bits needed to encode the next token given the previous context.

For SAIL, the variational framework adds a latent variable $z$, resulting in an upper bound on compression length through the ELBO:

$$\text{Compression Length of } G \leq -\log_2(p(G|z)) + D_{\text{KL}}(q(z \mid G) \parallel p(z)) \tag{8}$$

$$= -\sum_{t=1}^{T} \log_2(p_\theta(x_t|x_{<t}, z)) + D_{\text{KL}} \tag{9}$$

The KL divergence term is computed as follows:

$$D_{\text{KL}}(q(z \mid G) \parallel p(z)) = \frac{1}{2}\sum_{i=1}^{d} \left(\mu_i^2 + \sigma_i^2 - 1 - \log(\sigma_i^2)\right) \cdot \log_2(e) \tag{10}$$

where $d$ is the latent dimensionality and the factor $\log_2(e)$ converts from nats to bits. The autoregressive formulation naturally handles variable-length graphs through the sequential factorization, eliminating the need for separate structure and entity terms.

This provides an upper bound on the true compression length; the VAE's ELBO is a lower bound on log-likelihood, which, when negated, becomes an upper bound on compression. The bound is particularly relevant as the autoregressive decoder must account for uncertainty in token ordering during generation.

### A.3.3 SAMPLING FROM LATENT VARIABLE, $z$

We conduct two types of generation experiments:

1. *Unconditional Generation:* We sample 10,000 random latent codes from the standard normal prior distribution $p(z) = \mathcal{N}(0, I)$ and decode them into complete graphs using beam search with beam width $k = 3$. Each decoded graph is analyzed for: (1) semantic validity according to dataset-specific constraints, (2) novelty by checking against the training and validation sets, and (3) non-emptiness to ensure the model generates meaningful structures rather than null graphs.

2. *Conditional Generation:* We evaluate the model's ability to complete partial graphs by providing incomplete sequences as prompts. For each test graph, we provide the first $n$ tokens (*e.g.*, $[\texttt{BOS}, h_1, r_1, t_1]$) and generate the remaining sequence autoregressively. We vary the conditioning length and measure: (1) the semantic validity of the completed graph and (2) the diversity of completions when sampling with different random seeds.

### A.3.4 INTERPOLATION IN LATENT SPACE

We conduct both quantitative and qualitative analyses of the latent space structure:

1. *Quantitative Analysis:* We measure latent space smoothness through four metrics: (1) *Local Smoothness* – average Jaccard similarity between consecutive decoded graphs along random walks in latent space with step size $\epsilon = 0.1$, measuring whether small movements produce similar graphs; (2) *Global Consistency* – Jaccard similarity between each step and the anchor point, measuring drift from the starting graph; (3) *Flip Rate* – fraction of steps that produce different decoded graphs, with lower rates indicating larger basins of attraction in latent space; and (4) *Average Basin Length* – mean number of consecutive interpolation steps that decode to identical graphs, quantifying the granularity of the learned representation. For each metric, we sample multiple anchor points and random directions, taking 10-30 steps along each trajectory.

2. *Qualitative Analysis:* We visualize the latent space structure using two approaches: (1) *2D Projection* – we encode all test graphs and project their latent representations to 2D using t-SNE, coloring points by semantic attributes (genre for wd-movies) to observe clustering patterns; and (2) *Linear Interpolation* – we select pairs of semantically distinct graphs, encode them to obtain $z_1$ and $z_2$, then decode intermediate points $z_\alpha = (1 - \alpha)z_1 + \alpha z_2$ for $\alpha \in [0, 1]$ at regular intervals to examine the semantic coherence of interpolated graphs.

### A.4 QUALITATIVE ANALYSIS OF CONDITIONAL SAMPLING

**Qualitative Results**    We test whether SAIL has learned meaningful latent representations that capture director-specific collaborative patterns and genre preferences, despite never being explicitly trained on individual directorial styles. Figure A.1 shows representative examples of conditional generation for director-specific movie graphs. When conditioned on "Tim Burton" as the director, the model successfully generates graphs featuring his frequent collaborators (Helena Bonham Carter, Christopher Lee) and characteristic genres (Comedy Film, Musical Film). SAIL captures Burton's tendency to work repeatedly with the same ensemble cast, demonstrating learned patterns of directorial collaboration. In contrast, the Wes Anderson generation fails to capture his distinctive style. This disparity in generation quality likely reflects differences in dataset representation; Burton's more frequent appearances and consistent casting patterns in the training data enabled better pattern learning, while Anderson's style may have been underrepresented. Despite these variations

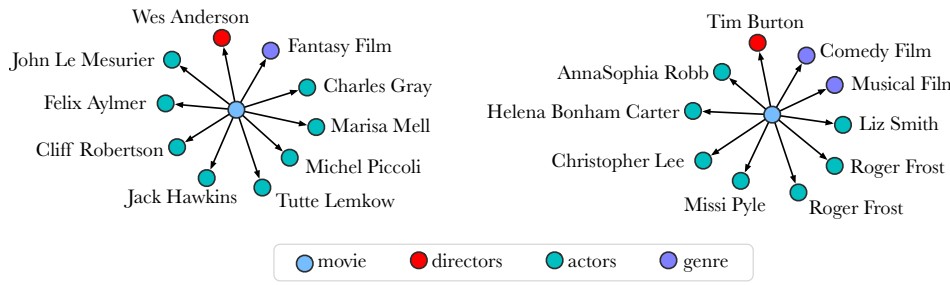

Figure A.1: Graphs generated by ARK conditioned on director entities for Wes Anderson **(left)** and (b) Tim Burton **(right)**. Node colors indicate entity types: movie (blue), directors (red), actors (green), and genres (purple).

in director-specific accuracy, both generated graphs maintain semantic validity as movie KGs, indicating that the SAIL has learned general graph structure.

### A.5 ABLATION STUDY

We systematically analyze the contribution of key architectural components through two ablation experiments on the syn-paths dataset, examining both model capacity and architectural choices.

**Method** We conduct two complementary ablation studies:

1. *Architectural Hyperparameter Analysis:* We vary the number of GRU layers $n_{\text{layers}} \in \{1, 2, 3, 4, 5\}$ and model dimensions $d_{\text{model}} \in \{2, 4, 8, 16, 32, 64, 128, 256, 512\}$ while keeping other hyperparameters fixed. For each configuration, we train the model until convergence and evaluate generation by measuring the percentage of semantically valid and novel graphs. We also test the relative importance of network depth versus hidden dimensionality on generation quality.

2. *Architecture Ablation:* We systematically replace transformer components with simpler architectures to assess their contribution: (1) *MLP Encoder* – replaces the transformer encoder with a multi-layer perceptron while preserving positional encoding; (2) *GRU Decoder* – replaces the transformer decoder with a GRU-based sequential decoder; and (3) *MLP Encoder & GRU Decoder* – combines both modifications, using an MLP encoder and GRU decoder. Each variant maintains comparable parameter counts to the transformer baseline for fair comparison.

**Architectural Hyperparameter Analysis Results** In Figure A.2, the model dimension has a substantially stronger impact on generation quality than network depth. Varying the number of layers from 1 to 5 produces relatively stable performance around 45% valid & novel rate, though with high variance across configurations. In contrast, the center panel demonstrates a sharp performance threshold: models with fewer than 16 hidden units achieve near-zero validity rates, while those with $d_{\text{model}} \geq 64$ consistently achieve 70-95% validity. The right panel's scatter plot confirms this pattern across individual runs, showing clear stratification by model dimension rather than layer count (indicated by color). These findings suggest that for KG generation on syn-paths dataset, a single-layer GRU with sufficient hidden units ($\geq 64$) can match or exceed the performance of deeper networks, supporting our claim that architectural simplicity does not compromise generation quality when coupled with appropriate capacity.

**Architecture Ablation Results** To better understand the contribution of architectural choices, we compare our full transformer-based model $t$-SAIL against simplified variants: SAIL, which replaces the transformer encoder and decoder with an MLP encoder and a GRU decoder, an MLP encoder (paired with a transformer decoder), $t$-ARK, a decoder-only transformer model, and ARK, a GRU decoder-only model. These ablations allow us to isolate the effect of transformer components in both the encoder and the decoder, and to assess whether an encoder is required for KG generation at all. In addition to generation quality and compression efficiency, we also report relative training time, as computational efficiency is often a limiting factor in scaling generative models. Table

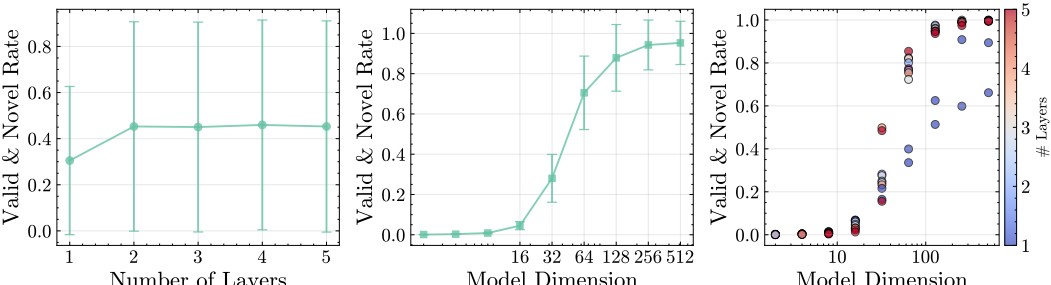

Figure A.2: Effect of architectural hyperparameters on the semantic validity and novelty. **(Left)** Valid & Novel rate as a function of the number of GRU layers, showing stable performance across depths with high variance. **(Center)** Performance variation with model dimension (hidden units), demonstrating a sharp improvement threshold around 64 dimensions, followed by consistent high performance. **(Right)** Scatter plot of individual experimental runs showing the relationship between model dimension and generation quality, with color indicating the number of layers.

3 demonstrates that transformer components, while improving generation quality, are not strictly necessary for effective knowledge graph modeling. Sequential decoders are consistently the most efficient: ARK trains at **0.09–0.27**,× the baseline time (i.e., **3.7–11×** faster) with near baseline validity across datasets, and its sequential inductive bias is competitive for decoding *e.g.*, syn-tipr (23.48 bits, on par with $t$-ARK 's 23.34) and wd-movies (**98.19** bits, best overall). Meanwhile, SAIL yields the best compression on wd-articles ( **199.55** bits), indicating that modest latent structure plus a GRU decoder can improve efficiency on complex, real-world graphs. Taken together, these results suggest that, for KG generation, a strong sequential decoder often dominates architectural choice, and the extra cost of full transformers, especially in the decoder, may be hard to justify when compute is constrained.

## A.6   CONDITIONED GENERATION

Figure A.3 shows that conditioned generation is also possible for the ARK model, which allows the model to generate KGs and simultaneously enforces specific constraints. Entities or relations are fixed in place in the positions of interest, and then we decode the remaing tokens with constrained sampling (temperature/top-k/top-p). Figure A.3a shows that the novelty and validity of the generated structures remain high for all steps of the conditioning process, an indication that the model can produce triples and, consequently, graphs that are semantically correct. At the same time, as seen in Figure A.3b, the diversity of the generated graphs drops dynamically as more entities and relations are added. This makes sense as the population of probable samples narrows with each additional constraint and limits the generative freedom of the model.

## A.7   ADDITIONAL COMMENTS ABOUT ARK & SAIL

**Variable Graph Length**     It is desirable to learn latent graph structures of varying sizes. In natural language processing, language models utilize special tokens to indicate the end of a sequence. Following a similar approach, we model variable length KGs by linearizing graphs into sequence of tokens and intoducing boundary tokens. We always introduce `BOS` as the inital token and terminate generation upon emitting `EOS`, while using `PAD` for mini batching. This simple setting allows the decoder to learn *when* to stop and *how large* the generated graphs should be, ensuring that the length distribution is learned. During inference, beam search halts on `EOS`, leading to the production of graphs of different sizes without any post hoc trim. In order to avoid length bias, we randomize triple order during training. In the probabilistic variant (SAIL), the latent **z** conditions the entire sequence and this yields consistent length control across all samples, while at the same time preserving variability.

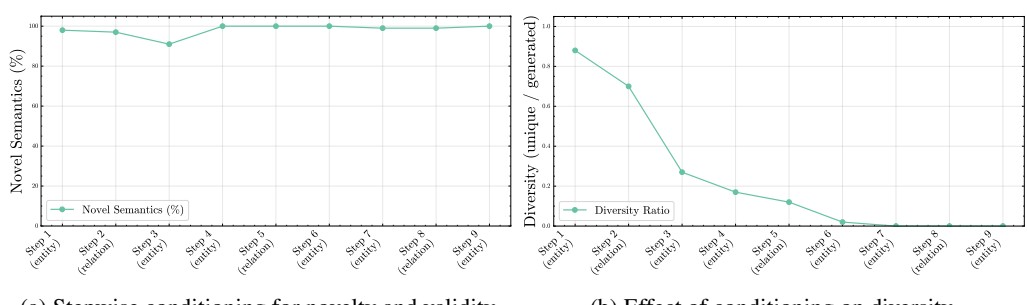

(a) Stepwise conditioning for novelty and validity.

(b) Effect of conditioning on diversity.

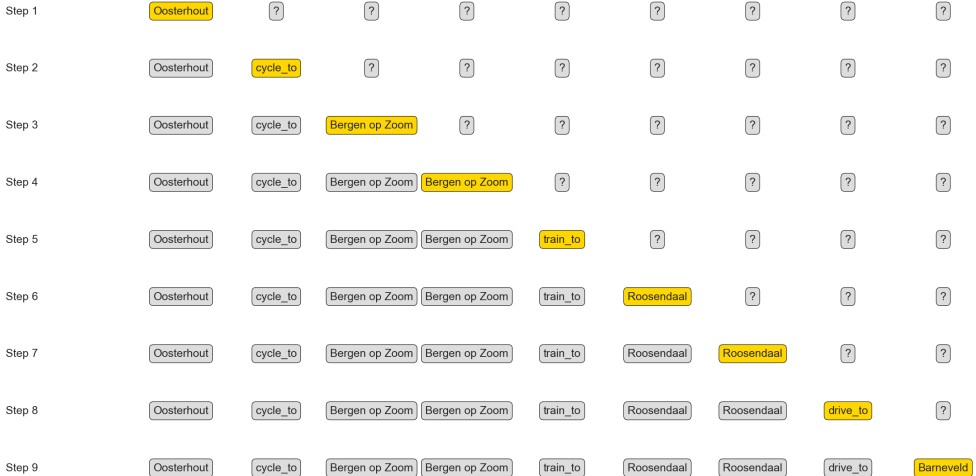

(c) Example of syn-paths conditioned generation

Figure A.3: Effect of progressive conditioning on Knowledge Graph generation for the syn-paths dataset. Subfigure (a) quantifies novelty and validity under increasing conditioning, (b) shows the corresponding reduction in sample diversity, and (c) provides an example of a conditioned generation where the model completes a partially specified graph.

| Datasets | Model | % Valid Generation ↑ | % Novel Graphs ↑ | Compression (bits) ↓ | Training Time ↓ |
|---|---|---|---|---|---|
| **syn-paths** | $t$-SAIL | 99.60 | 100.00 | 27.77 | 1.00 |
| | SAIL | 92.50 | 100.00 | 28.74 | 0.21 |
| | MLP Encoder | 99.80 | 100.00 | 27.35 | 0.55 |
| | $t$-ARK | 97.39 | 100.00 | 27.57 | 0.12 |
| | ARK | 99.95 | 100.00 | 27.65 | 0.09 |
| **syn-tipr** | $t$-SAIL | 100.00 | 100.00 | 26.30 | 1.00 |
| | SAIL | 98.45 | 100.00 | 27.14 | 0.17 |
| | MLP Encoder | 99.48 | 100.00 | 26.30 | 0.20 |
| | $t$-ARK | 100.00 | 100.00 | 23.34 | 0.17 |
| | ARK | 100 | 100.00 | 23.48 | 0.09 |
| **syn-types** | $t$-SAIL | 100.00 | 100.00 | 59.61 | 1.00 |
| | SAIL | 100.00 | 100.00 | 60.58 | 0.39 |
| | MLP Encoder | 93.27 | 100.00 | 59.33 | 0.41 |
| | $t$-ARK | 87.07 | 100.00 | 59.79 | 0.18 |
| | ARK | 89.22 | 100.00 | 59.63 | 0.09 |
| **wd-movies** | $t$-SAIL | 99.83 | 100.00 | 124.50 | 1.00 |
| | SAIL | 99.47 | 100.00 | 116.84 | 0.24 |
| | MLP Encoder | 99.44 | 100.00 | 118.64 | 0.36 |
| | $t$-ARK | 98.33 | 100.00 | 114.49 | 0.23 |
| | ARK | 99.24 | 100.00 | 98.19 | 0.21 |
| **wd-articles** | $t$-SAIL | 98.00 | 96.00 | 235.24 | 1.00 |
| | SAIL | 99.13 | 100.00 | 199.55 | 0.42 |
| | MLP Encoder | 97.7 | 100.00 | 206.23 | 0.48 |
| | $t$-ARK | 95.37 | 100.00 | 224.25 | 0.33 |
| | ARK | 97.24 | 100.00 | 205.24 | 0.27 |

Table 3: Architectural ablation study comparing ARK against simplified architectures with MLP encoders and GRU decoders. We evaluate model variants across five datasets using generation quality metrics (percentage of valid and novel graphs), compression efficiency (bits required for latent representation), and computational efficiency (training time relative to $t$-SAIL baseline).

## B  ADDITIONAL TABLES

| Datasets | Model | % Valid Graphs ↑ | % Novel & Valid ↑ | % Novel Graphs ↑ | % Empty Graphs ↓ |
|---|---|---|---|---|---|
| | uniform | 0 | 0 | 100.00 | 0 |
| | TransE | 0.25 | 0.25 | 23.45 | 76.55 |
| | DistMult | 0.69 | 0.69 | 14.59 | 85.41 |
| **syn-paths** | ComplEx | 0.71 | 0.71 | 14.27 | 85.73 |
| | $t$-SAIL | 99.60 | 99.60 | 100.00 | 0 |
| | SAIL | 92.50 | 92.50 | 100.00 | 0 |
| | $t$-ARK | 97.39 | 97.39 | 100.00 | 0 |
| | ARK | **99.95** | **99.95** | 100.00 | 0 |
| | uniform | 0 | 0 | 100.00 | 0 |
| | TransE | 0 | 0 | 5.58 | 94.42 |
| | DistMult | 0 | 0 | 13.34 | 86.66 |
| **syn-tipr** | ComplEx | 0 | 0 | 4.95 | 96.05 |
| | $t$-SAIL | 100.00 | 100.00 | 100.00 | 0 |
| | SAIL | 98.45 | 98.45 | 100.00 | 0 |
| | $t$-ARK | 100.00 | 100.00 | 100.00 | 0 |
| | ARK | **100.00** | **100.00** | 100.00 | 0 |
| | uniform | 0 | 0 | 100.00 | 0 |
| | TransE | 0.21 | 0.21 | 15.44 | 84.56 |
| | DistMult | 0.13 | 0.13 | 12.46 | 87.53 |
| **syn-types** | ComplEx | 0.07 | 0.07 | 10.25 | 89.75 |
| | $t$-SAIL | 100.00 | **100.00** | **100.00** | 0 |
| | SAIL | 100.00 | 100.00 | 100.00 | 0 |
| | $t$-ARK | 87.07 | 87.07 | 100.00 | 0 |
| | ARK | 89.22 | 89.22 | 100.00 | 0 |
| | uniform | 0 | 0 | 100.00 | 0 |
| | TransE | 0 | 0 | 14.61 | 85.39 |
| | DistMult | 0 | 0 | 12.93 | 87.07 |
| **wd-movies** | ComplEx | 0 | 0 | 1.87 | 98.13 |
| | $t$-SAIL | **99.83** | **99.9** | **100** | 0 |
| | SAIL | 99.47 | 99.47 | 100.00 | 0 |
| | $t$-ARK | 98.33 | 98.33 | 100.00 | 0 |
| | ARK | 99.24 | 99.24 | 100.00 | 0 |
| | uniform | 0 | 0 | 100.00 | 0 |
| | TransE | 0 | 0 | 4.58 | 95.42 |
| | DistMult | 0 | 0 | 0 | 100.00 |
| **wd-articles** | ComplEx | 0 | 0 | 2.46 | 97.54 |
| | $t$-SAIL | 98.00 | 98.00 | 100.00 | 0 |
| | SAIL | **99.13** | **99.13** | 100.00 | 0 |
| | $t$-ARK | 95.37 | 95.37 | 100.00 | 0 |
| | ARK | 97.24 | 97.24 | 99.99 | 0 |

Table 4: Semantic validity of the graphs generated. We sample graphs and check the novelty of the sampled graphs by comparing them against the training and validation sets. The best performing models for each dataset are **bolded**. Baseline results are from the IntelliGraphs paper (Thanapalasingam et al., 2023).

## C ADDITIONAL FIGURES

### C.1 ARCHITECTURAL DETAILS

Figure C.1 shows the architectural details of the $t$-SAIL model. Also Figure A.1 shows an example for conditioned generation.

| Datasets | Models | Compression Length (bits) | | | |
|---|---|---|---|---|---|
| | | $G$ | $S$ | $E$ | $D_{KL}$ |
| **syn-paths** | uniform | 30.49 | 12.80 | 17.69 | - |
| | TransE | 49.89 | 16.19 | 33.69 | - |
| | ComplEx | 54.39 | 20.71 | 33.69 | - |
| | DistMult | 48.58 | 14.90 | 33.69 | - |
| | $t$-SAIL | 27.77 | - | 14.47 | 13.30 |
| | SAIL | 28.74 | - | 18.41 | 10.33 |
| | $t$-ARK | **27.57** | - | - | - |
| | ARK | 27.65 | - | - | - |
| **syn-tipr** | uniform | 61.61 | 29.14 | 32.47 | - |
| | TransE | 69.51 | 28.70 | 40.81 | - |
| | ComplEx | 63.96 | 23.15 | 40.81 | - |
| | DistMult | 67.51 | 26.70 | 40.81 | - |
| | $t$-SAIL | 26.30 | - | 11.13 | 15.17 |
| | SAIL | 27.14 | - | 9.90 | 17.24 |
| | $t$-ARK | **23.34** | - | - | - |
| | ARK | 23.48 | - | - | - |
| **syn-types** | uniform | **36.02** | 16.84 | 19.18 | - |
| | TransE | 48.26 | 19.05 | 29.21 | - |
| | ComplEx | 47.69 | 18.48 | 29.21 | - |
| | DistMult | 47.46 | 18.24 | 29.21 | - |
| | $t$-SAIL | 59.61 | - | 59.46 | 0.15 |
| | SAIL | 60.58 | - | 60.37 | 0.21 |
| | $t$-ARK | 59.79 | - | - | - |
| | ARK | 59.63 | - | - | - |
| **wd-movies** | uniform | 171.60 | 53.86 | 117.74 | - |
| | TransE | 208.60 | 51.39 | 157.21 | - |
| | ComplEx | 202.68 | 45.46 | 157.21 | - |
| | DistMult | 208.50 | 51.29 | 157.21 | - |
| | $t$-SAIL | 124.50 | - | 92.66 | 31.84 |
| | SAIL | 116.84 | - | 100.10 | 16.74 |
| | $t$-ARK | 114.49 | - | - | - |
| | ARK | **98.19** | - | - | - |
| **wd-articles** | uniform | 693.80 | 295.60 | 398.20 | - |
| | TransE | 910.65 | 280.67 | 629.98 | - |
| | ComplEx | 887.30 | 257.33 | 629.98 | - |
| | DistMult | 901.91 | 271.94 | 629.98 | - |
| | $t$-SAIL | 235.24 | - | 225.60 | 9.64 |
| | SAIL | **199.55** | - | 186.38 | 13.17 |
| | $t$-ARK | 224.25 | - | - | - |
| | ARK | 205.24 | - | - | - |

Table 5: We measure the compression quality for compressing graphs $G$. $D_{KL}$ is only available for the VAE because it relies on the variational approximation, which is unique to this model. For the VAE, we compute an upper bound on the compression length (in bits). Probabilistic baseline (uniform, TransE, ComplEx, DistMult) results are from Thanapalasingam et al. (2023).

| Dataset | Model | Local Smoothness ↑ | Global Consistency ↑ | Flip Rate ↓ | Avg Basin Length ↑ |
|---|---|---|---|---|---|
| **syn-paths** | $t$-SAIL | 0.75 | 0.36 | 0.20 | 4.54 |
| | SAIL | 0.74 | 0.14 | 0.33 | 2.87 |
| **syn-tipr** | $t$-SAIL | 0.99 | 0.98 | 0.09 | 8.61 |
| | SAIL | 0.93 | 0.69 | 0.10 | 8.03 |
| **syn-types** | $t$-SAIL | 0.82 | 0.60 | 0.12 | 6.80 |
| | SAIL | 0.92 | 0.73 | 0.20 | 4.47 |
| **wd-movies** | $t$-SAIL | 0.87 | 0.58 | 0.15 | 5.70 |
| | SAIL | 0.84 | 0.49 | 0.40 | 2.93 |
| **wd-articles** | $t$-SAIL | 0.81 | 0.55 | 0.14 | 5.37 |
| | SAIL | 0.82 | 0.57 | 0.17 | 3.46 |

Table 6: Latent space smoothness metrics for $t$-SAIL and SAIL with $\epsilon = 0.1$. Higher local/global smoothness indicates more continuous transitions. Lower flip rates suggest larger regions mapping to identical graphs.

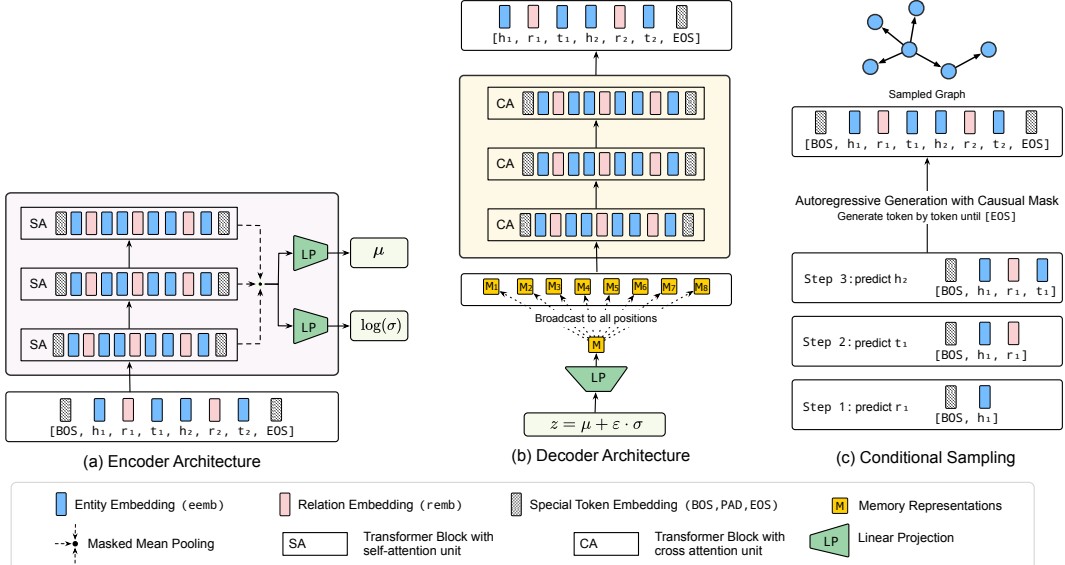

Figure C.1: $t$-SAIL has three main components: (a) an *Encoder* that processes linearized Knowledge Graph triple sequences $[\text{BOS}, h_1, r_1, t_1, h_2, r_2, t_2, \ldots, \text{EOS}]$ through self-attention (SA) blocks to produce latent distribution parameters $(\mu, \log \sigma)$, (b) a *Decoder* that uses cross-attention (CA) to condition on the sampled latent code $z$ and autoregressively generates token sequences with causal masking, and (c) *Conditional Sampling* that demonstrates the step-by-step autoregressive generation process, predicting one token at a time until the [EOS] token is produced or the maximum sequence length is reached. The model uses a unified vocabulary embedding matrix spanning special tokens ([BOS], [PAD], [EOS]), entities (shown in blue), and relations (shown in pink), enabling sequential generation of Knowledge Graphs from learned latent representations.