# OpenReview forum: "When Simplicity Wins: Efficient Knowledge Graph Generation with Sequential Decoders"
_ICLR.cc/2026/Conference — ICLR 2026 Conference Withdrawn Submission_

### Official Review · Reviewer_3iXx · 2025-10-21

**Soundness:** 2
**Presentation:** 2
**Contribution:** 2
**Rating:** 2
**Confidence:** 4

**Summary:**

This paper proposes two autoregressive models for KG generation: ARK, a GRU-based sequential decoder, and SAIL, a variational extension. The authors argue that these simple recurrent models can achieve competitive semantic validity and compression compared to transformer-based alternatives, at a fraction of the computational cost. Experiments on the IntelliGraphs benchmark show 89-100% "semantic validity" and up to 11x faster training.

**Strengths:**

- The authors highlight an important practical question—whether graph generation requires expensive attention mechanisms—and attempt to quantify the trade-off between complexity and performance.
- The reported efficiency gains are substantial and consistently measured.
- The results are presented cleanly, with clear numerical comparisons to transformer variants and baselines.

**Weaknesses:**

- The paper never clearly defines what the model is trained or evaluated to do. It oscillates between unconditional generation, conditional completion, and sampling from a prior, but the experiments appear to involve only unconditional sampling of new graphs composed of existing entities and relations. It is thus unclear what constitutes success or correctness. The task does not resemble realistic KG completion or open-world graph expansion.

- The model linearizes graphs into sequences of triples and trains with next-token prediction. This violates the fundamental permutation invariance of graphs: the likelihood $p(G)$ depends on the arbitrary order of triples. Randomizing triple order during training (as mentioned in Section 3.1) is a heuristic, not a principled solution. As a result, the model may learn the distribution over sequences, not graphs.

- Definition 2.2 lists semantic constraints as generic rules (e.g., type or temporal consistency) but provides no formalism, rule language, or enforcement mechanism. The notion of "semantic validity" is therefore dataset-specific and ill-defined. It could mean anything from simple type checks to arbitrary heuristics, making the reported validity scores difficult to interpret.

- The models have no mechanism for constraint satisfaction. There appears to be no constraint-aware loss, masked decoding, or logical regularization. `Semantic validity is only evaluated post-hoc by checking generated graphs against dataset rules. Thus, high validity numbers may reflect statistical memorization of patterns rather than any explicit reasoning or rule satisfaction.

- The IntelliGraphs benchmark treats small, disconnected subgraphs as independent samples and evaluates them on semantic validity, novelty, and compression. None of these metrics assess factual accuracy, plausibility, or usefulness for reasoning. Since the benchmark's constraints are synthetic and the datasets small, solving it may not indicate progress on real KG generation, where graphs are incomplete, open-world, and semantically rich.

- In SAIL, the prior $p(z)$ is a fixed standard Gaussian, not learned from data. The model's latent space is therefore only loosely regularized and may not correspond to meaningful graph semantics. Sampling from $p(z)$ does not constitute learning a structured generative model of KGs.

- The paper claims that ``simplicity wins,'' but the evidence is confined to a narrow and partially synthetic benchmark of limited realism. There is no evidence that the results generalize to realistic or large-scale knowledge graphs. The work would be more credible as a small-scale efficiency study rather than as a broad claim about model inductive biases.

**Questions:**

- What precisely is the generation task—graph completion, sampling, or reconstruction?
- How are semantic constraints represented and evaluated? Are they logical rules, templates, or dataset-specific checks?
- Is the model aware of these constraints during training, or are they only applied post hoc?
- How is permutation invariance handled beyond randomization?

---

### Official Review · Reviewer_WfZt · 2025-10-24

**Soundness:** 2
**Presentation:** 1
**Contribution:** 2
**Rating:** 4
**Confidence:** 4

**Summary:**

This paper introduces ARK (Auto-Regressive Knowledge Graph Generation), a sequential decoder framework for KG generation. Unlike prior approaches that rely on transformer-based attention-heavy architectures, ARK uses lightweight GRU-based recurrent models to sequentially generate triples as token sequences.

Key contributions for this paper are:
- Proposing ARK, A GRU-based autoregressive decoder that generates semantically valid KGs by modeling dependencies across triples.
- Proposing SAIL, An extension of ARK with a variational latent space, allowing controlled generation, interpolation, and diversity in KG generation.
- This paper shows great efficiency, as RNNs achieve comparable semantic validity (89.2–100%) and compression performance to transformers while requiring only 9–21% of training time.
- This paper provides empirical insights, such as model capacity (hidden size $\geq$ 64) matters more than depth; shallow GRUs match deep transformers.
- Extensive evaluation on IntelliGraphs benchmarks (synthetic + real-world datasets) shows ARK/SAIL outperform traditional KG embeddings and rival transformer baselines, while being computationally efficient

These results suggest that simplicity (RNNs) can outperform complex transformers for KG generation, overturning assumptions that attention mechanisms are essential for structured data generation.
Oveall, this paper is well-executed and insightful, with strong empirical support for its claim that simplicity can rival complexity in KG generation. Its main limitations concern writing quality, technical novelty, scalability, and open-world applicability.

**Strengths:**

- The paper challenges the widespread assumption that transformers are necessary for structured data generation, showing that GRUs can be equally or more effective.
- The paper proposes a sequential decoding perspective for KGs, framing them as token sequences of triples.
- The SAIL extension with variational inference introduces controllability in KG generation, a significant step for practical applications.
- Results could reshape how KG generation models are designed, encouraging efficient, lightweight models over resource-intensive transformers, and hopefully inspires reconsideration of simplicity vs. complexity tradeoffs in other structured generative tasks.

**Weaknesses:**

- Only tested on IntelliGraphs (synthetic + Wikidata-derived). While useful, these datasets are relatively small (graphs up to 212 triples). It remains unclear how the models scale to very large or heterogeneous KGs (e.g., enterprise-scale KGs).
- Vocabulary of entities and relations is fixed at training time. This limits open-world applicability, where new entities emerge dynamically. Future-proofing with compositional or inductive representations would strengthen the work.
- By linearizing KGs into sequences, the model imposes an arbitrary ordering on inherently unordered graphs. Although randomization mitigates bias, it may still affect generalization.
- While SAIL adds latent controllability, practical use cases (e.g., controllable interpolation beyond synthetic transitions) are underexplored. The semantic plausibility of interpolated graphs is not deeply validated (e.g., realistic KG completion).
- The technical novelty of this architecture is limited. Core ideas (GRU-based autoregression, variational latent extension) are adaptations of established techniques. The novelty lies more in the application and empirical insight than in methodological innovation.

**Questions:**

- How does ARK/SAIL perform on much larger graphs (e.g., >10k triples) beyond the tested benchmarks? Do GRUs retain efficiency and semantic validity under extreme sequence lengths?
- Have you evaluated whether different linearization strategies (e.g., BFS vs. DFS traversal of the graph) influence generation quality?
- Can ARK/SAIL be extended to inductive settings (e.g., using entity embeddings learned unseen, on-the-fly, or compositional representations)?
- While interpolations show smoothness, how well does the latent space correspond to interpretable KG properties (e.g., temporal progression, ontology consistency)? Could disentanglement improve controllability?
- Do you envision ARK/SAIL being deployed in real-world KG completion systems (e.g., Wikidata augmentation)? If so, how would you mitigate issues of bias, factual errors, and scalability mentioned in the limitations?
- The authors had better attach the Appendix directly after References, not put them at Supplementary Material, which is hard to follow.

---

### Official Review · Reviewer_4g4Y · 2025-10-28

**Soundness:** 3
**Presentation:** 3
**Contribution:** 2
**Rating:** 6
**Confidence:** 3

**Summary:**

This paper introduces ARK, a family of autoregressive models for knowledge graph (KG) generation, demonstrating that simple GRU-based sequential decoders can match or even outperform transformer-based models in both efficiency and quality. The authors challenge the prevailing assumption that attention mechanisms are essential for structured data generation and back their claims with extensive experiments on synthetic and real-world datasets. They also propose SAIL, a variational extension of ARK, to support controlled diversity in KG generation.

**Strengths:**

- The paper proposes some autoregressive models for KG generation, which is a new but practical task.

- The paper shows some interesting findings. For example, single-layer GRUs with sufficient hidden dimensionality (≥64) can match deep transformer models in KG generation tasks. On IntelliGraphs benchmarks, the GRU variant achieves 89.2–100.0% semantic validity, with only <0.76% degradation compared to transformers.

**Weaknesses:**

- In my opinion, the technical contribution of this paper is limited. The authors formulate the KG generation task as a sequence generation problem and build their method upon existing sequential models. From this perspective, the paper lacks originality in its technical methods. Moreover, it does not clearly articulate the inherent challenges of the KG generation task itself.

- Although the paper presents extensive experimental data and comparisons, the methods used are all fairly basic. Many other sequence modeling architectures, especially those involving large generative models such as KGT5, could potentially be applied to this task but have not been evaluated.

**Questions:**

- The task is called KG generation, but it seems to only generate subgraphs composed of known entities and relations. Is it unable to generate new entities or new relations?

- What is the motivation for choosing ARK and VAE? Can a standard RNN perform this task as well? What are the technical challenges of this task?

---

### Official Review · Reviewer_sBSk · 2025-10-28

**Soundness:** 2
**Presentation:** 2
**Contribution:** 1
**Rating:** 4
**Confidence:** 4

**Summary:**

This paper proposes ARK for generating semantically valid knowledge graphs by treating KGs as sequences of triples. The authors argue that despite the structured nature of KGs, simple sequential decoders can match or even outperform Transformer-based counterparts in terms of semantic validity. Experiments on the IntelliGraphs dataset demonstrate promising results of the proposed method.

**Strengths:**

The paper presents a well-defined, end-to-end autoregressive approach to KG generation, with explicit linearization, training, and decoding procedures.

ARK and SAIL achieve near-perfect semantic validity on synthetic tasks and strong performance on real-world subsets.

**Weaknesses:**

The paper evaluates only on small graphs (3–212 triples), which are far from real-world KG scales. It is unclear whether the proposed sequential approach scales to larger, denser, or more diverse graphs.

Comparing against KGE models (TransE, DistMult) as generative baselines is misleading. These models were never designed for joint graph generation. They score triples independently and lack mechanisms to sample coherent subgraphs. In fact, there are multiple works exploring the generation of KG triples, e.g., [1–2].

Using RNNs to model triple sequences has already been explored [3]. The authors overstate the novelty of using RNNs for sequence generation. Also, it is confusing, as Transformers also generate sequences autoregressively (e.g., in LLMs).

Given the rise of LLMs for structured data generation, the paper’s exclusive focus on small GRUs/Transformers feels outdated.

Treating KGs as sequences ignores their inherent unordered, set-like nature. While the authors randomize triple order during training, this does not fully address permutation sensitivity. By the way, is it more reasonable to generate BFS/DFS paths [3]?

[1] Start from zero: Triple set prediction for automatic knowledge graph completion. TKDE 2024.

[2] Revisit and outstrip entity alignment: A perspective of generative models. ICLR 2024.

[3] Learning to exploit long-term relational dependencies in knowledge graphs. ICML 2019.

**Questions:**

Please see Weaknesses.

---

### Note · Authors · 2025-11-21

**Comment:**

Thank you to the reviewers for their thoughtful feedback. After careful consideration, we believe this work would be better suited for a different venue and have decided to withdraw our submission.

**Withdrawal Confirmation:**

I have read and agree with the venue's withdrawal policy on behalf of myself and my co-authors.